# Labor prior to cesarean delivery associated with higher post-discharge opioid consumption

**Holly B. Ende**[1☯*], **Ruth Landau**[2‡], **Naida M. Cole**[3‡], **Sara M. Burns**[4☯], **Brian T. Bateman**[3☯], **Melissa E. Bauer**[5‡], **Jessica L. Booth**[6‡], **Pamela Flood**[7‡], **Lisa R. Leffert**[4‡], **Timothy T. Houle**[4‡], **Lawrence C. Tsen**[3☯]

**1** Department of Anesthesiology, Vanderbilt University Medical Center, Nashville, Tennessee, United States of America, **2** Department of Anesthesiology, Columbia University Irving Medical Center, New York, New York, United States of America, **3** Department of Anesthesiology, Brigham and Women's Hospital, Harvard Medical School, Boston, Massachusetts, United States of America, **4** Department of Anesthesiology, Critical Care, and Pain Medicine, Massachusetts General Hospital, Harvard Medical School, Boston, Massachusetts, United States of America, **5** Department of Anesthesiology, Division of Obstetric Anesthesiology, University of Michigan Medical School, Ann Arbor, Michigan, United States of America, **6** Department of Anesthesiology, Wake Forest School of Medicine, Winston-Salem, North Carolina, United States of America, **7** Department of Anesthesiology, Perioperative and Pain Medicine, Stanford University, Stanford, California, United States of America

☯ These authors contributed equally to this work.
‡ These authors also contributed equally to this work.
\* holly.ende@vumc.org

**Data Availability Statement:** Data cannot be shared publicly because of concerns of patient confidentiality regarding sharing sensitive survey

## Abstract

### Background

Severe acute post-cesarean delivery (CD) pain has been associated with an increased risk for persistent pain and postpartum depression. Identification of women at increased risk for pain can be used to optimize post-cesarean analgesia. The impact of labor prior to CD (intrapartum CD) on acute post-operative pain and opioid use is unclear. We hypothesized that intrapartum CD, which has been associated with both increased inflammation and affective distress related to an unexpected surgical procedure, would result in higher postoperative pain scores and increased opioid intake.

### Methods

This is a secondary analysis of a prospective cohort study examining opioid use up to 2 weeks following CD. Women undergoing CD at six academic medical centers in the United States 9/2014-3/2016 were contacted by phone two weeks following discharge. Participants completed a structured interview that included questions about postoperative pain scores and opioid utilization. They were asked to retrospectively estimate their maximal pain score on an 11-point numeric rating scale at multiple time points, including day of surgery, during hospitalization, immediately after discharge, 1st week, and 2nd week following discharge.

Pain scores over time were assessed utilizing a generalized linear mixed-effects model with the patient identifier being a random effect, adjusting for an a priori defined set of con-founders. A multivariate negative binomial model was utilized to assess the association

data on pain and opioid use. Additionally, public release of the data was not discussed with patients during the consent process. Requests for data can be forwarded to: Massachusetts General Hospital Anesthesia Research Center c/o: Ariel Mueller 55 Fruit Street, White 5 Boston, Massachusetts 02114 Email: almueller@mgh.harvard.edu Phone: 617-726-9252.

**Funding:** The authors received no specific funding for this work.

**Competing interests:** The authors have declared that no competing interests exist.

between intrapartum CD and opioid utilization after discharge, also adjusting for the same confounders. In the context of non-random prescription distribution, this model was constructed with an offset for the number of tablets dispensed.

## Results

A total of 720 women were enrolled, 392 with and 328 without labor prior to CD. Patients with intrapartum CD were younger, less likely to undergo repeat CD or additional surgical procedures, and more likely to experience a complication of CD.

Women with intrapartum CD consumed more opioid tablets following discharge than women without labor (median 20, IQR 10–30 versus 17, IQR 6–30; p = 0.005). This association persisted after adjustment for confounders (incidence rate ratio 1.16, 95% CI 1.05–1.29; p = 0.004). Pain scores on the day of surgery were higher in women with intrapartum CD (difference 0.91, 95% CI 0.52–1.30; adj. p = <0.001) even after adjustment for confounders. Pain scores at other time points were not meaningfully different between the two groups.

## Conclusion

Intrapartum CD is associated with worse pain on the day of surgery but not other time points. Opioid requirements following discharge were modestly increased following intrapartum CD.

## Introduction

Accounting for approximately 32% of deliveries in 2015 [1], cesarean delivery (CD) rates in the United States are likely to continue to rise given the increasing maternal age and prevalence of obesity and diabetes [2]. Pain following CD is typically managed effectively with multimodal analgesia; however, when poorly treated, chronic pain and depression may occur [3].

Acute postsurgical pain has been associated with patient (e.g., age, preexisting psychiatric diagnoses, body mass index, smoking status), surgical (e.g., planned versus unplanned CD, primary versus repeat CD, operative time), and anesthetic (general versus regional anesthesia) factors [4–9]. Pre-procedural identification of women at increased risk for pain following CD can be used to optimize analgesia and potentially attenuate development of chronic effects [10]. The impact of labor prior to CD on acute post-operative pain scores and post-discharge opioid consumption is unclear. We hypothesized that intrapartum CD, which has been associated with both increased inflammation [11] and affective distress related to an unexpected CD [12], would result in higher postoperative pain scores and increased opioid intake compared to CD without prior labor (unlabored CD).

## Methods and methods

### Subject enrollment and consent

This is a secondary analysis of a prospective cohort study evaluating patterns of opioid use after CD [13]. The survey study was performed on women who underwent unlabored and intrapartum CD at six academic medical centers in the United States between September 2014 and March 2016. Participants represented a convenience sample of women who had undergone cesarean delivery at each center and were subsequently identified by the study team by

daily rounding, review of the electronic medical record, or notification by the obstetric team. All women post-cesarean delivery were considered for inclusion unless one of the following exclusion criteria were met: limited English proficiency, lack of capacity to provide consent, age <18 years, or hospital length of stay >7 days following delivery. Centers included Brigham and Women's Hospital (Boston, MA), Massachusetts General Hospital (Boston, MA), the University of Michigan (Ann Arbor, MI), Columbia University Medical Center (New York, NY), Wake Forest Health Science Center (Winston-Salem, NC), and Stanford University Medical Center (Palo Alto, CA). Institutional Review Board (IRB) approval was obtained at all participating institutions.

Consent procedures varied by institution based on the requirements imposed by the IRB at each participation center. At Brigham and Women's Hospital, Massachusetts General Hospital, and the University of Michigan, research staff or members of the patient's care team approached potential subjects during their hospitalization with information about the study. Women were informed that approximately two weeks after delivery they would be contacted by a study team member to participate in an interview pertaining to their postoperative experience and were advised that they could opt out at that time or any subsequent time via phone or email. At the time of the interview two weeks following discharge, verbal consent was obtained from all subjects by the primary investigator or a co-investigator prior to proceeding with the interview. Written consent was not required by the IRB at these centers. Participant consent was documented in the REDCap (Research Electronic Data Capture) database used for data collection. At Stanford University Medical Center, Wake Forest Health Sciences Center, and Columbia University Medical Center, subjects provided written informed consent prior to hospital discharge. This manuscript adheres to the applicable CONSORT guidelines.

## Data collection

Collected survey data were transcribed into REDCap Database, a secure, web-based application designed for research studies [14]. The presence or absence of labor prior to CD was identified by manual chart review. The data represent a convenience sample, with the number of subjects contributed by each center determined by logistical factors. Subject enrollment, consent, and survey methods are as previously described in detail by Bateman et al [13]. In brief, patients were surveyed two weeks following hospital discharge via a structured phone interview (structured interview guide available via https://cdn-links.lww.com/permalink/aog/a/aog_130_1_2017_05_02_bateman_17-212_sdc1.pdf) which included questions pertaining to current pain experience, opioid and other analgesic use, medication-related side effects, and overall satisfaction with pain management. The patients were asked to recall maximum pain scores on the day of surgery, during hospitalization, at time of discharge, and 1- and 2-weeks post discharge. Information regarding opioid prescriptions was obtained verbally from patients or, when not available, from the subject's medical record.

## Statistical analysis

The primary outcome of interest was the number of opioid tablets consumed following discharge, with the exposure of interest being intrapartum CD. A multivariate negative binomial model was utilized to test for this effect, adjusting for an *a priori* defined set of covariates which included maternal age, race, insurance type, repeat delivery, multiple gestation, tubal ligation, surgical complications, length of stay following delivery, history of smoking, history of alcohol abuse, history of drug abuse, antidepressant use, benzodiazepine use, non-steroidal anti-inflammatory drug (NSAID) use, anesthesia type, opioid type, and center where delivery occurred. In the context of non-random prescription distribution, the model was constructed

with an offset for the number of tablets dispensed. As a secondary outcome of interest, verbal analog scale (VAS) pain scores over time were assessed utilizing a generalized linear mixed-effects model, with the patient identifier being a random effect. Pain at rest at the time of the 2-week survey was assessed separately utilizing a negative binomial model. Results are reported as incident rate ratios (IRR) or least-squares means with 95% confidence intervals (CI). P-values are adjusted using the Bonferroni correction for multiple comparisons.

Baseline subject demographics and characteristics are reported as frequency count (%), mean±SD, or median [25th, 75th], as appropriate. Differences between patients who did and did not experience labor prior to CD were compared using a Wilcoxon rank sum test, chi-squared test, or Fisher's Exact test, as appropriate. All analyses were conducted using R statistical software (RStudio, version 3.2.2; R Foundation for Statistical Computing, Vienna, Austria), and all hypothesis testing was two-tailed.

## Results

A total of 1065 parturients were considered for inclusion in this study, with 720 included in the final analysis (Fig 1), of which 328 (45.5%) experienced intrapartum CD and 392 (54.5%) did not. The parturients were primarily white (59.2%), had private insurance (76.9%), were non-smokers (86.4%) with no self-reported abuse history of alcohol (94.4%) or illicit substances (95.7%). The majority did not take antidepressants (93.8%) or benzodiazepines (99.0%) during pregnancy. Differences in baseline characteristics between the two groups are shown in Table 1. Patients with intrapartum CD were younger, less likely to have had a prior CD or to be undergoing additional surgical procedures, and more likely to experience a complication of CD and have an indwelling epidural catheter that was used to provide cesarean anesthesia. Patients without prior labor were more likely to undergo a CD under spinal anesthesia. Of those patients undergoing intrapartum CD, 52/328 (16%) were repeat CD, indicating a failed trial of labor after CD (TOLAC).

Overall, the median number of opioid tablets dispensed was 40 [IQR 30 to 40] in both groups (Table 2). The majority of prescriptions were for oxycodone or oxycodone with

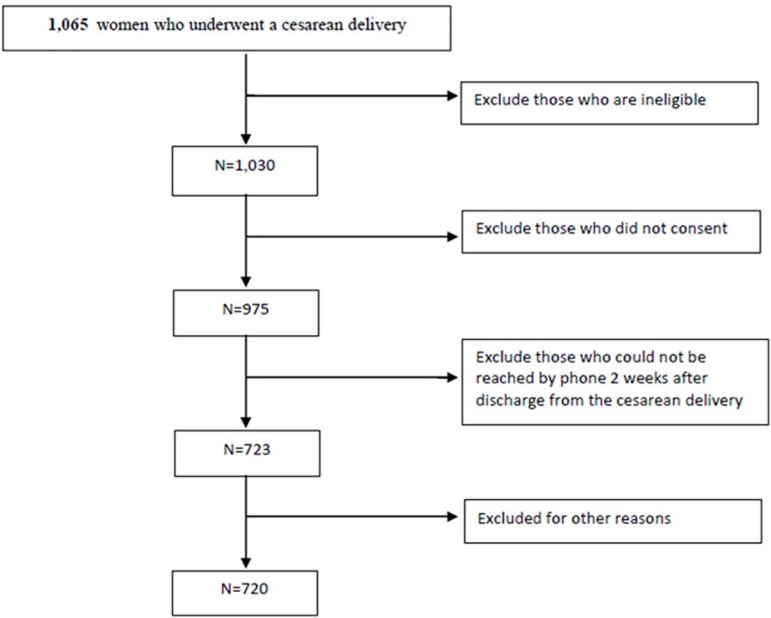

**Fig 1. Patient flow chart.**

**Table 1.  Characteristics of patients who labored or did not labor prior to cesarean delivery.**

| | Labor Prior to CD (n = 328) | No Labor Prior to CD (n = 392) | p-value |
|---|---|---|---|
| **Age (years)** | 32±6 | 33±5 | 0.001 |
| **Race** | | | 0.642 |
| White | 200 (61.0) | 226 (57.9) | |
| Black | 49 (14.9) | 60 (15.4) | |
| Hispanic | 31 (9.5) | 48 (12.3) | |
| Asian/Pacific Islander | 22 (6.7) | 32 (8.2) | |
| Other | 5 (1.5) | 3 (0.8) | |
| Unknown | 21 (6.4) | 21 (5.4) | |
| **Insurance Type** | | | 0.930 |
| Private | 258 (78.7) | 296 (75.9) | |
| Medicaid | 62 (18.9) | 84 (21.5) | |
| Unknown | 3 (0.9) | 4 (1.0) | |
| Other | 3 (0.9) | 4 (1.0) | |
| None | 2 (0.6) | 2 (0.5) | |
| **Type of CD** | | | <0.001 |
| Primary | 276 (84.1) | 174 (44.4) | |
| Repeat | 52 (15.9) | 218 (55.6) | |
| **Additional Surgical Procedure** | | | |
| None | 282 (86.0) | 300 (76.5) | 0.002 |
| Closed rectus abdominus | 4 (1.2) | 5 (1.3) | 0.999 |
| Hysterectomy | 1 (0.3) | 2 (0.5) | 0.999 |
| Tubal ligation | 22 (6.7) | 65 (16.6) | <0.001 |
| Other | 22 (6.7) | 28 (7.1) | 0.935 |
| **Complication of CD** | 41 (12.5) | 13 (3.3) | <0.001 |
| **Total hospital length of stay** | 4 [4, 5] | 4 [3, 4] | <0.001 |
| **Smoking Status** | | | 0.670 |
| Smoker | 46 (14.0) | 52 (13.3) | |
| Non-smoker | 282 (86.0) | 340 (86.7) | |
| **History of alcohol abuse** | | | 0.405 |
| Yes | 15 (4.6) | 14 (3.6) | |
| No | 310 (94.8) | 370 (94.9) | |
| Unknown | 2 (0.6) | 6 (1.5) | |
| **History of substance abuse** | | | 0.192 |
| Yes | 13 (4.0) | 7 (1.8) | |
| No | 311 (95.1) | 378 (96.9) | |
| Unknown | 3 (0.9) | 5 (1.3) | |
| **Preoperative Antidepressant Use** | 20 (6.1) | 25 (6.4) | 0.999 |
| **Preoperative Benzodiazepine Use** | 4 (1.2) | 3 (0.8) | 0.812 |
| **Postoperative NSAID use** | 309 (94.2) | 364 (92.9) | 0.563 |
| **Anesthesia Type** | | | |
| Spinal | 75 (22.9) | 292 (74.5) | <0.001 |
| Epidural | 175 (53.4) | 12 (3.1) | <0.001 |
| Combined spinal-epidural[a] | 76 (23.2) | 82 (20.9) | 0.524 |
| General | 7 (2.1) | 5 (1.3) | 0.546 |
| **Study Site** | | | 0.006 |
| Brigham and Women's Hospital | 82 (25.0) | 117 (29.8) | |
| Columbia University Medical Center | 73 (22.3) | 99 (25.3) | |

*(Continued)*

**Table 1.** (Continued)

|  | **Labor Prior to CD (n = 328)** | **No Labor Prior to CD (n = 392)** | **p-value** |
|---|---|---|---|
| **Massachusetts General Hospital** | 88 (26.8) | 108 (27.6) |  |
| **Stanford University Medical Center** | 3 (0.9) | 4 (1.0) |  |
| **University of Michigan Medical Center** | 37 (11.3) | 15 (3.8) |  |
| **Wake Forest University Medical Center** | 45 (13.7) | 49 (12.5) |  |

[a]The combined spinal-epidural category includes both those performed during labor (with epidural catheter bolused for cesarean delivery) and those performed intraoperatively

All data presented as mean±sd, median[IQR], or n(%)

CD-cesarean delivery, NSAID-nonsteroidal anti-inflammatory drug

acetaminophen, and mean number of tablets dispensed varied by institution. The primary outcome of interest was the number of opioid tablets consumed after discharge. To account for patient characteristics, we used a negative binomial model to examine the association between intrapartum CD and the number of tablets consumed, while adjusting for potential confounders. An association between labor prior to CD and opioid consumption post-discharge persisted after adjustment for these confounders (IRR 1.16, 95% CI 1.05 to 1.29, p = 0.004), with women undergoing intrapartum CD consuming an adjusted incidence rate of 16% more opioid tablets compared to women with no prior labor (Table 2). Secondary outcomes included VAS pain scores on the day of delivery, during hospitalization, at time of discharge and 1- and 2-weeks post discharge. VAS pain scores were consistently higher in women with an intrapartum CD at all time points, but the difference was significant on the day of delivery only (difference 0.91, 95% CI 0.52 to 1.30, adj. p = <0.001, Fig 2).

## Discussion

In the current study, we observed that women with an intrapartum CD reported higher pain scores on the day of surgery and consumed 16% more opioids tablets (median 17 vs 20) following hospital discharge, than women undergoing a CD without prior labor. These women were younger and less likely to be undergoing a repeat CD, in this case a failed trial of labor after previous CD, or an additional surgical procedure (e,g. bilateral tubal ligation), but more likely to have transitioned from labor epidural analgesia and to have experienced a post-cesarean complication (eg. postoperative bleeding or infection).

Greater opioid consumption after an intrapartum CD, compared with that after unlabored CD, has been previously observed. In a recent, single center, retrospective study, Prabhu et al. observed greater opioid consumption 72 to 96 hours postoperatively in parturients with >75[th] percentile and >90[th] percentile of all opioid consumption after an unscheduled compared to scheduled CD. This difference was not observed at other timepoints during the delivery hospitalization [4]. Duration of labor may also correlate with post-CD opioid consumption, as suggested in a recent study examining women who underwent intrapartum CD [15]. There are many potential reasons that intrapartum CD may be associated with greater pain and opioid

**Table 2. Summary of opioid tablets dispensed, consumed, and leftover for women who labored or did not labor prior to cesarean delivery.**

|  | **Labor prior to cesarean** | **No labor prior to cesarean** | **p-value** |
|---|---|---|---|
| **Tablets dispensed (median[IQR])** | 40 [30, 40] | 40 [30, 40] | 0.253 |
| **Tablets consumed (median[IQR])** | 20 [10, 30] | 17 [6, 30] | 0.005 |

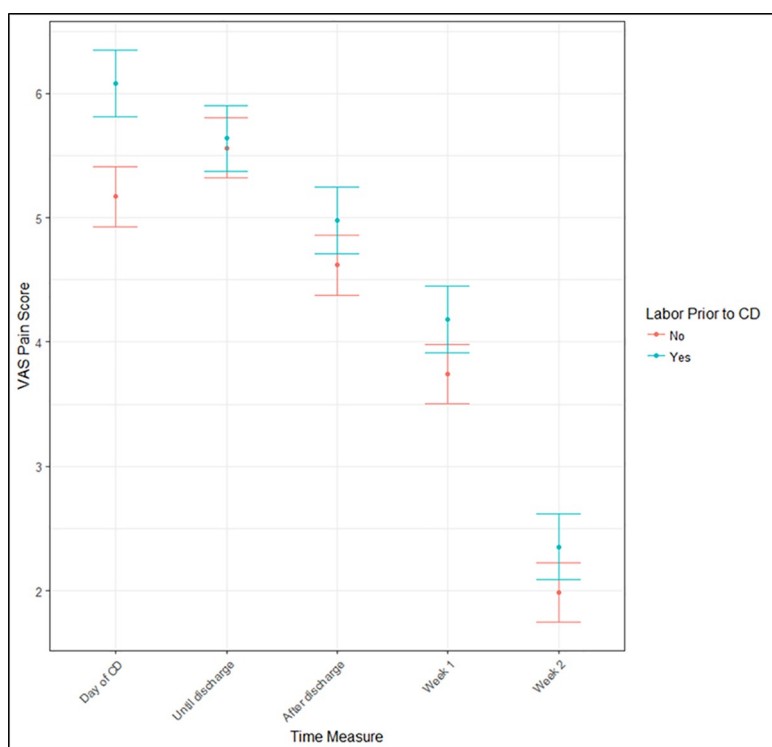

**Fig 2. Median pain scores over time of women who labored prior to cesarean delivery versus those who did not.**

consumption. It is likely that laboring women expecting a vaginal delivery were less emotionally or psychologically prepared for major abdominal surgery. Moreover, poor preoperative sleep quality, as may be experienced during labor, may contribute to worsened pain postoperatively [16]. In addition, a change in maternal or fetal status, particularly if abrupt and sufficient to warrant an operative delivery, would likely enhance anxiety. The presence of preoperative anxiety and fear, and undergoing an emergent compared to elective CD, have been associated with higher postoperative pain scores, opioid consumption, anhedonia and depression [4, 17–20]. Moreover, women intent on a vaginal delivery who undergo an intrapartum CD can experience a postpartum sense of personal failure, which can negatively impact the birth experience and recovery [21]. Surgical technique during urgent or emergent CD may also contribute to increased postoperative pain, even in the absence of documented complications. Lastly, intrapartum CD by definition includes all TOLACs. It is unclear how failed TOLAC status may impact pain and opioid consumption; however, a recent publication demonstrated lower median oxycodone dose with failed TOLAC compared to primary intrapartum CD [15].

Our findings corroborate the observation that even modest differences in postoperative pain scores, in our study a 0.91 point difference on the day of surgery, can identify patients who may develop persistent pain. Eisenach, et al., demonstrated that for each point increase in acute pain following vaginal or CD, there was an associated 12.7% increase in persistent pain and an 8.3% increase in depressive symptoms at 8 weeks postpartum [3]. Furthermore, Komatsu, et al., found that pain scores on postoperative day 1 and 2 were predictive of time to pain resolution and time to opioid cessation [22]. While the women in our study who underwent an intrapartum CD were younger and more likely to experience surgical complications, both of which may be associated with more severe acute pain, the association persisted even after adjusting for these confounders [23]. The difference in pain scores translated clinically to

higher opioid intake following discharge in these women. Identification of intrapartum CD as a risk factor for greater pain and opioid consumption postoperatively can bring attention to factors associated with labor (eg. inflammation, psychological trauma) which can be targeted to limit the amount of opioid needed post-delivery (eg. standing nonsteroidal anti-inflammatory medications, postpartum trauma counseling).

There are important limitations to this study. The current investigation was performed as a post-hoc analysis of a prior study [13]; thus, an *a priori* power calculation was not conducted. However, given the large sample size, it was possible to detect small, significant differences in outcomes. Additionally, the patients included in the original survey study represented a convenience sample of women undergoing CD at six academic medical centers across the United States, which may not be representative of other practice settings. More specifically, participants were primarily white, privately insured, nonsmokers, with very low rates of illicit substance use; all these elements could limit the generalizability of these results. In addition, a standard 40 tablet opioid prescription is no longer common following CD, and as the number of opioid tablets consumed is related to the number of tablets dispensed, the median opioid consumption in our study likely overestimates current realities [13]. Postpartum women were contacted via telephone two weeks following delivery; therefore, loss to follow-up and recall bias are additional limitations. Approximately 25% of the women consented for our study could not be reached at 2 weeks following delivery (252/975). Recall bias was likely limited for the quantification of consumed opioid tablets as patients were asked to count leftover tablets; however, the retrospective estimation of pain scores by subjects may limit the interpretation of these results. While anesthetic techniques differed significantly between groups, with intrapartum CD more commonly utilizing a pre-sited epidural catheter and unlabored CD more commonly performed under spinal anesthesia, the reported differences in pain scores and opioid consumption persisted after controlling for these factors.

In conclusion, women with an intrapartum CD have modestly higher opioid consumption in the first two weeks after delivery. Since judicious opioid prescribing is an important strategy to reduce the likelihood for persistent opioid use, labor prior to CD should be factored into models aiming to predict the optimal amount of opioids prescribed after CD.

## Author Contributions

**Conceptualization:** Holly B. Ende, Ruth Landau, Naida M. Cole, Brian T. Bateman, Melissa E. Bauer, Jessica L. Booth, Pamela Flood, Lisa R. Leffert, Timothy T. Houle, Lawrence C. Tsen.

**Data curation:** Sara M. Burns, Timothy T. Houle.

**Formal analysis:** Holly B. Ende, Sara M. Burns, Timothy T. Houle.

**Investigation:** Holly B. Ende, Brian T. Bateman, Lawrence C. Tsen.

**Methodology:** Holly B. Ende, Brian T. Bateman.

**Project administration:** Holly B. Ende.

**Supervision:** Brian T. Bateman, Lawrence C. Tsen.

**Writing – original draft:** Holly B. Ende, Brian T. Bateman, Lawrence C. Tsen.

**Writing – review & editing:** Holly B. Ende, Ruth Landau, Naida M. Cole, Sara M. Burns, Brian T. Bateman, Melissa E. Bauer, Jessica L. Booth, Pamela Flood, Lisa R. Leffert, Timothy T. Houle.

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
