## [Decision Letter · Decision Letter 0]

17 Mar 2021

PONE-D-20-40071

Labor prior to cesarean delivery associated with higher post-discharge opioid consumption

PLOS ONE

Dear Dr. Ende,

Thank you for submitting your manuscript to PLOS ONE. After careful consideration, we feel that it has merit but does not fully meet PLOS ONE’s publication criteria as it currently stands. Therefore, we invite you to submit a revised version of the manuscript that addresses the points raised during the review process.

Please revise accordingly.

We look forward to receiving your revised manuscript.

Kind regards,

Academic Editor

PLOS ONE

Journal Requirements:

2. Please provide additional details regarding participant consent. In the ethics statement in the Methods and online submission information, please describe how verbal consent was documented and witnessed, and why written consent was not obtained.

3. In your Methods section, please provide additional information about the participant recruitment method and the demographic details of your participants. Please ensure you have provided sufficient details to replicate the analyses such as:

a) a description of any inclusion/exclusion criteria that were applied to participant recruitment

b) a statement as to whether your sample can be considered representative of a larger population

4. Please include a copy of the structured interview guide used in the study, in both the original language and English as Supporting Information, or include a citation if it has been published previously.

5. We noted in your submission details that a portion of your manuscript may have been presented or published elsewhere.

"This is a secondary analysis of a prospective cohort study evaluating patterns of opioid use after cesarean delivery (Bateman BT, et al. Patterns of Opioid Prescription and Use After Cesarean Delivery. Obstet Gynecol. 2017;130(1):29-35). Original publication is attached as Related Manuscript File."

Please clarify whether this publication was peer-reviewed and formally published. If this work was previously peer-reviewed and published, in the cover letter please provide the reason that this work does not constitute dual publication and should be included in the current manuscript.

6. We note that you have indicated that data from this study are available upon request. PLOS only allows data to be available upon request if there are legal or ethical restrictions on sharing data publicly. For information on unacceptable data access restrictions, please see http://journals.plos.org/plosone/s/data-availability#loc-unacceptable-data-access-restrictions.

7. Thank you for submitting the above manuscript to PLOS ONE. During our internal evaluation of the manuscript, we found that your abstract appears to have been previously published with the Society for Obstetric Anesthesia and Perinatology. Before we proceed, would you please kindly clarify if the published extended abstract was previously peer-reviewed? If so, please explain in your cover letter why this work does not constitute a dual publication. Please also clarify at this time whether your abstract was previously copyrighted.

Reviewers' comments:

Reviewer's Responses to Questions

**Comments to the Author**

1. Is the manuscript technically sound, and do the data support the conclusions?

Reviewer #1: Yes

Reviewer #2: Yes

2. Has the statistical analysis been performed appropriately and rigorously? 

Reviewer #1: Yes

Reviewer #2: I Don't Know

3. Have the authors made all data underlying the findings in their manuscript fully available?

Reviewer #1: Yes

Reviewer #2: Yes

4. Is the manuscript presented in an intelligible fashion and written in standard English?

Reviewer #1: Yes

Reviewer #2: Yes

5. Review Comments to the Author

Reviewer #1: Thank you for your work. The results are consistent with the research hypothesis. The discussion covers the analysis of the results with comparison with other articles. tables are clear. I recommend publishing the paper

Reviewer #2: ABSTRACT: There are just few grammatical errors in the abstract that need to be corrected.

METHODS: Why were these 6 centers chosen for the study and why did some of the participants give a written consent and a verbal consent obtained from the others (page 5 lines 114 and 116) Apart from that this secondary analysis of a previous prospective cohort study is acceptable for publication.

6. PLOS authors have the option to publish the peer review history of their article (what does this mean?). If published, this will include your full peer review and any attached files.

Reviewer #1: **Yes: **Ahmed M. Abbas

Reviewer #2: No

---

## [Author Response · Author response to Decision Letter 0]

14 May 2021

General Comments:

Comment 1: Please ensure that your manuscript meets PLOS ONE's style requirements, including those for file naming. The PLOS ONE style templates can be found at

Response 1: We confirm that we have reviewed the PLOS ONE style requirements to ensure this manuscript meets those criteria.

Comment 2: Please provide additional details regarding participant consent. In the ethics statement in the Methods and online submission information, please describe how verbal consent was documented and witnessed, and why written consent was not obtained.

Response 2: This additional detail has been added to the Methods section, under the subheading of Subject enrollment and consent – “…verbal consent was obtained from all subjects by the primary investigator or a co-investigator prior to proceeding with the interview. Written consent was not required by the IRB at these centers. Participant consent was documented in the REDCap (Research Electronic Data Capture) database used for data collection.”

Comment 3: In your Methods section, please provide additional information about the participant recruitment method and the demographic details of your participants. Please ensure you have provided sufficient details to replicate the analyses such as:

a) a description of any inclusion/exclusion criteria that were applied to participant recruitment

b) a statement as to whether your sample can be considered representative of a larger population

Response 3: These additional details have been added to the Methods section, under the subheading of Subject enrollment and consent – “Participants represented a convenience sample of women who had undergone either elective or unplanned cesarean delivery at each center and were subsequently identified by the study team by daily rounding, review of the electronic medical record, or notification by the obstetric team. All women post-cesarean delivery were considered for inclusion unless one of the following exclusion criteria were met: limited English proficiency, lack of capacity to provide consent, age <18 years, or hospital length of stay >7 days following delivery.” A statement on generalizability and representativeness of our sample is provided in the Discussion section – “the patients included in the original survey study represented a convenience sample of women undergoing CD at six academic medical centers across the United States, which may not be representative of other practice settings.” 

Comment 4: Please include a copy of the structured interview guide used in the study, in both the original language and English as Supporting Information, or include a citation if it has been published previously.

Response 4: The hyperlink to the structured interview guide has added to the Methods section, under the subheading of Data Collection.

Comment 5: We noted in your submission details that a portion of your manuscript may have been presented or published elsewhere. "This is a secondary analysis of a prospective cohort study evaluating patterns of opioid use after cesarean delivery (Bateman BT, et al. Patterns of Opioid Prescription and Use After Cesarean Delivery. Obstet Gynecol. 2017;130(1):29-35). Original publication is attached as Related Manuscript File." Please clarify whether this publication was peer-reviewed and formally published. If this work was previously peer-reviewed and published, in the cover letter please provide the reason that this work does not constitute dual publication and should be included in the current manuscript.

Response 5: The original publication “Patterns of Opioid Prescription and Use After Cesarean Delivery” was peer-reviewed and published in Obstetrics & Gynecology in 2017. The current study is a secondary analysis of the data collected for that previous study, but represents a completely novel hypothesis, data analysis, and reporting of results. No outcomes or results from the prior study are re-presented in the current work. This detail has been added to the updated Cover Letter.

Comment 6a: If there are ethical or legal restrictions on sharing a de-identified data set, please explain them in detail (e.g., data contain potentially identifying or sensitive patient information) and who has imposed them (e.g., an ethics committee). Please also provide contact information for a data access committee, ethics committee, or other institutional body to which data requests may be sent. If there are no restrictions, please upload the minimal anonymized data set necessary to replicate your study findings as either Supporting Information files or to a stable, public repository and provide us with the relevant URLs, DOIs, or accession numbers.

Response 6a: This data represents sensitive patient information involving self-reported patient opioid use and pain scores. As we did not request permission for release of the de-identified dataset in the Institutional Review Board applications at all of the institutions included in the study, it is not ethically appropriate to release this sensitive information, even in deidentified form. Furthermore, while restrictions are not specifically imposed by any organization, public release of the data was not discussed with patients during the consent process and thus we cannot release patient-level data. 

Requests for data can be forwarded to: 

Massachusetts General Hospital Anesthesia Research Center 

c/o: Ariel Mueller

55 Fruit Street, White 5

Boston, Massachusetts 02114

Email: almueller@mgh.harvard.edu

Phone: 617-726-9252

Comment 7: During our internal evaluation of the manuscript, we found that your abstract appears to have been previously published with the Society for Obstetric Anesthesia and Perinatology. Before we proceed, would you please kindly clarify if the published extended abstract was previously peer-reviewed? If so, please explain in your cover letter why this work does not constitute a dual publication. Please also clarify at this time whether your abstract was previously copyrighted.

Response 7: The current work was presented in abstract form at the Society for Obstetric Anesthesia and Perinatology Annual Meeting in 2017. The abstract was not peer-reviewed or published. It was included only as part of the Annual Meeting Syllabus. Neither the abstract nor the current manuscript are copyrighted in any way. 

Reviewer 1: 

Comment 1: Thank you for your work. The results are consistent with the research hypothesis. The discussion covers the analysis of the results with comparison with other articles. tables are clear. I recommend publishing the paper

Response 1: Thank you for taking the time to review our manuscript and for this kind review of our paper. 

Reviewer 2: 

Comment 1: ABSTRACT: There are just few grammatical errors in the abstract that need to be corrected.

Response 1: Thank you for bringing our attention to any grammatical errors. We have edited punctuation on lines 61 and 78 of the Abstract.

Comment 2: METHODS: Why were these 6 centers chosen for the study and why did some of the participants give a written consent and a verbal consent obtained from the others (page 5 lines 114 and 116) 

Response 2: Additional detail has been added to the Methods section regarding consent procedures at the varying institutions. The six centers were chosen to represent a sampling of tertiary academic medical centers throughout the United States, in varying geographic regions.

Comment 3: Apart from that this secondary analysis of a previous prospective cohort study is acceptable for publication.

Response 3: Thank you for taking the time to review our manuscript and for your positive reception of our work.

Additional Requests from Editorial Office Email dated 5/7/21

Comment 1: Please indicate the policy, ethics committee, Institutional Review Board, or other organization that is imposing the restriction.

Response 1: While restrictions are not specifically imposed by any organization, public release of the data was not discussed with patients during the consent process and thus we cannot publicly release patient-level data.

Comment 2: Provide a non-author, institutional point of contact (and contact information) that is able to field data access queries. PLOS ONE's data policy requires this in the interest of maintaining long-term data accessibility.

Response 2: Requests for data can be forwarded to: 

Massachusetts General Hospital Anesthesia Research Center 

c/o: Ariel Mueller

55 Fruit Street, White 5

Boston, Massachusetts 02114

Email: almueller@mgh.harvard.edu

Phone: 617-726-9252

Comment 3: If relevant, add any data set names, variables, accession codes, URLs, DOIs, etc. that an independent researcher would need in order to request your minimal data set.

Response 3: Not applicable. Please see responses to comments 1 and 2 above. 

Comment 4: Thank you for providing the following response to our previous dual publication query: "The original publication “Patterns of Opioid Prescription and Use After Cesarean Delivery” was peer-reviewed and published in Obstetrics & Gynecology in 2017. The current study is a secondary analysis of the data collected for that previous study, but represents a completely novel hypothesis, data analysis, and reporting of results. No outcomes or results from the prior study are re-presented in the current work. This detail has been added to the updated Cover Letter." Before we proceed, would you please kindly clarify the following points:

Did the authors present any new data in this submission that were not previously presented in the published article Bateman BT, et al.? Did the authors perform any additional experiments or collect any additional data that were not a part of the study from the published article Bateman BT, et al.?

Response 4: The current work represents a new analysis of the data which were previously collected for the study published by Bateman BT, et al. For the previously published study, women were surveyed and reported their pain scores and number of opioid tablets consumed for two weeks following cesarean delivery. The study reported the median number of dispensed opioid tablets and the number of leftover tablets at the time of interview. In the current work, we tested the hypothesis that the number of opioid tablets consumed post-cesarean delivery would be greater in those patients who underwent intrapartum, as opposed to scheduled, cesarean delivery. This represents a new hypothesis, new data analysis, and reporting of new results not previously published in the 2017 article by Bateman BT, et al.

Comment 5: If the authors are conducting a "secondary analysis of the data collected for that previous study", please clarify whether the authors have the rights to both reuse and republish this data under a CC BY 4.0 license.

Response 5: We confirm that the data collected is owned by the investigators and that the authors have the rights to both reuse and republish the data under a CC BY 4.0 license.

---

## [Decision Letter · Decision Letter 1]

26 May 2021

PONE-D-20-40071R1

Labor prior to cesarean delivery associated with higher post-discharge opioid consumption

PLOS ONE

Dear Dr. Ende,

Thank you for submitting your manuscript to PLOS ONE. After careful consideration, we feel that it has merit but does not fully meet PLOS ONE’s publication criteria as it currently stands. Therefore, we invite you to submit a revised version of the manuscript that addresses the points raised during the review process.

Please revise accordingly.

We look forward to receiving your revised manuscript.

Kind regards,

Academic Editor

PLOS ONE

Journal Requirements:

Reviewers' comments:

Reviewer's Responses to Questions

**Comments to the Author**

1. If the authors have adequately addressed your comments raised in a previous round of review and you feel that this manuscript is now acceptable for publication, you may indicate that here to bypass the “Comments to the Author” section, enter your conflict of interest statement in the “Confidential to Editor” section, and submit your "Accept" recommendation.

Reviewer #3: All comments have been addressed

Reviewer #4: All comments have been addressed

2. Is the manuscript technically sound, and do the data support the conclusions?

Reviewer #3: Yes

Reviewer #4: Yes

3. Has the statistical analysis been performed appropriately and rigorously? 

Reviewer #3: Yes

Reviewer #4: Yes

4. Have the authors made all data underlying the findings in their manuscript fully available?

Reviewer #3: Yes

Reviewer #4: Yes

5. Is the manuscript presented in an intelligible fashion and written in standard English?

Reviewer #3: Yes

Reviewer #4: Yes

6. Review Comments to the Author

Reviewer #3: To authors,

The study was well-structured. The paper is well written. The authors faithfully reacted to the reviewer’s and editor’s comments. The paper has become markedly improved.

I have only one concern: I believe that in this study unplanned intrapartum CS vs. planned before-labor CS has been compared, right? All the context tells me so.

Let us sort out the terminology of cesarean section from two viewpoints: planned vs. unplanned and labor + vs. -. Naturally, there are four categories. 1) planned labor (+): breech presentation planned for CS tomorrow but night before labor occurred. Waiting three hours (she had pains every 10 minutes at CS), she had cesarean section at the planned data/time. This may be an extreme but we sometimes encounter the case in which “just before planned cesarean section, we found every ten minutes uterine contractions”. This apparently is defined as after labor (labor initiation is defined as ever ten minutes contraction with progressing cervical findings/ripening; the latter cannot be confirmed due to having done CS, thus every ten minutes contraction is sufficient to define “labor +”). 2) planned labor (-): no need to explain. 3) unplanned (or emergent) labor (+): failure to progress. 4) unplanned labor -; cardiotocography revealed non-reassuring fetal status requiring emergent CS. The context strongly suggest that you here compared 2) vs. 3).

Here, you used many terminologies, planned, unplanned, elective, non-elective, emergent, scheduled, unscheduled, labor (+), no-labor, intrapartum, etc. As an experienced obstetrician (4.5-decade of practice), I understand the situation; however, what is labor? What is planned? What is emergent? Please definitely state their criteria. For example, labor +/- might depend only presence/absence of every-ten-minute-contraction, right? As I described, one cannot confirm the cervical change and whether “this contraction” led to eventual delivery (not false pains) because some patients underwent CS. Thus, one may use the terminology “labor” under incomplete definition/criteria. Thus, please definitely define these terminologies and state what vs. what was actually compared here. I am not an English-native and thus I feel how wonderful it would be if you had used “consistent terminology”; planned CS vs. scheduled CS is “identical”, right? If so, please use the consistent terminology for the readers. However, if for USA (English native) people/doctors, the present terminology (variety of words) may be better, you can “ignore” my comment/advice. Sorry to write long.

For your convenience, I write parts that might cause confusion (due to terminology). From this viewpoint, there are many parts with mixed terminology also in discussion section. Do previous article (data) adopt the same criteria of labor + vs. -? Please confirm it. You need not write long.

You wrote:

Introduction last: We hypothesized that intrapartum CD, which has been associated with both increased inflammation[11] and affective distress related to an unexpected CD,[12] would result in higher postoperative pain scores and increased opioid intake compared to scheduled CD without prior labor.

Line 81: The survey study was performed on women who underwent scheduled and intrapartum CD at six academic medical centers in the United States between September 2014 and March 2016. Participants represented a convenience sample of women who had undergone either elective or unplanned cesarean delivery.

Line 197: Greater opioid consumption after an intrapartum CD, compared with that after scheduled CD, has been previously observed

Line 255: In conclusion, women with an unplanned intrapartum CD have modestly higher opioid consumption in the first 2 weeks after delivery.

Reviewer #4: The manuscript seems to be well-written and previous review comments are sound and the authors responded accordingly. I have a comment about the exclusion criteria. Isn't it better to exclude patients with endometriosis, uterine fibroid, and/or adenomyosis? The might influence on the postpartum pain. I would like to know about excluding young patients less than 18. Aged patients might also be probable confounder.

7. PLOS authors have the option to publish the peer review history of their article (what does this mean?). If published, this will include your full peer review and any attached files.

Reviewer #3: No

Reviewer #4: No

---

## [Author Response · Author response to Decision Letter 1]

2 Jun 2021

Reviewer 3

Comment 1: The study was well-structured. The paper is well written. The authors faithfully reacted to the reviewer’s and editor’s comments. The paper has become markedly improved. I have only one concern: I believe that in this study unplanned intrapartum CS vs. planned before-labor CS has been compared, right? All the context tells me so. Let us sort out the terminology of cesarean section from two viewpoints: planned vs. unplanned and labor + vs. -. Naturally, there are four categories. 1) planned labor (+): breech presentation planned for CS tomorrow but night before labor occurred. Waiting three hours (she had pains every 10 minutes at CS), she had cesarean section at the planned data/time. This may be an extreme but we sometimes encounter the case in which “just before planned cesarean section, we found every ten minutes uterine contractions”. This apparently is defined as after labor (labor initiation is defined as ever ten minutes contraction with progressing cervical findings/ripening; the latter cannot be confirmed due to having done CS, thus every ten minutes contraction is sufficient to define “labor +”). 2) planned labor (-): no need to explain. 3) unplanned (or emergent) labor (+): failure to progress. 4) unplanned labor -; cardiotocography revealed non-reassuring fetal status requiring emergent CS. The context strongly suggest that you here compared 2) vs. 3).

Response 1: Here we are comparing intrapartum CD (#1 and #3) versus CD without prior labor (#2 and #4). In order to improve transparency, we have adjusted the wording throughout the manuscript in various locations, as you’ve suggested below. The presence or absence of labor was determined by chart review by a study coinvestigator, so we cannot exclude the possibility of rare instances where infrequent contractions prior to a scheduled delivery were not captured in our study (#1 above).

Comment 2: Here, you used many terminologies, planned, unplanned, elective, non-elective, emergent, scheduled, unscheduled, labor (+), no-labor, intrapartum, etc. As an experienced obstetrician (4.5-decade of practice), I understand the situation; however, what is labor? What is planned? What is emergent? Please definitely state their criteria. For example, labor +/- might depend only presence/absence of every-ten-minute-contraction, right? As I described, one cannot confirm the cervical change and whether “this contraction” led to eventual delivery (not false pains) because some patients underwent CS. Thus, one may use the terminology “labor” under incomplete definition/criteria. Thus, please definitely define these terminologies and state what vs. what was actually compared here. I am not an English-native and thus I feel how wonderful it would be if you had used “consistent terminology”; planned CS vs. scheduled CS is “identical”, right? If so, please use the consistent terminology for the readers. However, if for USA (English native) people/doctors, the present terminology (variety of words) may be better, you can “ignore” my comment/advice. Sorry to write long.

Response 2: We have adjusted the wording in the manuscript to use consistent terminology (intrapartum vs. unlabored). All mentions of planned, unplanned, elective, non-elective, emergent, scheduled, and unscheduled have been removed from the manuscript (unless they describe the results of other studies).

Comment 3: For your convenience, I write parts that might cause confusion (due to terminology). From this viewpoint, there are many parts with mixed terminology also in discussion section. Do previous article (data) adopt the same criteria of labor + vs. -? Please confirm it. You need not write long.

Response 3: See responses to below comments. The terminology used to describe the results of prior articles is consistent with the methods and results described by those studies (ex. planned vs. unplanned or emergent vs. elective is used when that is how they were described in prior studies). 

Comment 4: Introduction last: We hypothesized that intrapartum CD, which has been associated with both increased inflammation[11] and affective distress related to an unexpected CD,[12] would result in higher postoperative pain scores and increased opioid intake compared to scheduled CD without prior labor.

Response 4: The wording has been adjusted for consistency – “We hypothesized that intrapartum CD, which has been associated with both increased inflammation[11] and affective distress related to an unexpected CD,[12] would result in higher postoperative pain scores and increased opioid intake compared to CD without prior labor (unlabored CD).”

Comment 5: Line 81: The survey study was performed on women who underwent scheduled and intrapartum CD at six academic medical centers in the United States between September 2014 and March 2016. Participants represented a convenience sample of women who had undergone either elective or unplanned cesarean delivery…

Response 5: The wording has been adjusted for consistency – “The survey study was performed on women who underwent unlabored and intrapartum CD at six academic medical centers in the United States between September 2014 and March 2016. Participants represented a convenience sample of women who had undergone cesarean delivery…”

Comment 6: Line 197: Greater opioid consumption after an intrapartum CD, compared with that after scheduled CD, has been previously observed

Response 6: The wording has been adjusted for consistency – “Greater opioid consumption after an intrapartum CD, compared with that after unlabored CD, has been previously observed.”

Comment 7: Line 255: In conclusion, women with an unplanned intrapartum CD have modestly higher opioid consumption in the first 2 weeks after delivery.

Response 7: The wording has been adjusted for consistency – “In conclusion, women with an intrapartum CD have modestly higher opioid consumption in the first two weeks after delivery.”

Reviewer 4 

Comment 1: The manuscript seems to be well-written and previous review comments are sound and the authors responded accordingly. I have a comment about the exclusion criteria. Isn't it better to exclude patients with endometriosis, uterine fibroid, and/or adenomyosis? The might influence on the postpartum pain. 

Response 1: While each of these comorbidities may influence postpartum pain and opioid consumption, unfortunately we did not collect data on or account for these conditions. Future work should likely take these conditions into account in verifying our study’s results.

Comment 2: I would like to know about excluding young patients less than 18. Aged patients might also be probable confounder.

Response 2: Patients under the age of 18 were excluded due to constraints of subject consent of minors. While there was a statistically significant difference in maternal age between the two groups (Table 1), age was included as a possible confounder in the multivariate analysis, and so differences in age should not be the cause of our reported outcomes.

---

## [Decision Letter · Decision Letter 2]

17 Jun 2021

Labor prior to cesarean delivery associated with higher post-discharge opioid consumption

PONE-D-20-40071R2

Dear Dr. Ende,

We’re pleased to inform you that your manuscript has been judged scientifically suitable for publication and will be formally accepted for publication once it meets all outstanding technical requirements.

Kind regards,

Academic Editor

PLOS ONE

Additional Editor Comments (optional):

Reviewers' comments:

Reviewer's Responses to Questions

**Comments to the Author**

1. If the authors have adequately addressed your comments raised in a previous round of review and you feel that this manuscript is now acceptable for publication, you may indicate that here to bypass the “Comments to the Author” section, enter your conflict of interest statement in the “Confidential to Editor” section, and submit your "Accept" recommendation.

Reviewer #3: All comments have been addressed

Reviewer #4: All comments have been addressed

2. Is the manuscript technically sound, and do the data support the conclusions?

Reviewer #3: Yes

Reviewer #4: Yes

3. Has the statistical analysis been performed appropriately and rigorously? 

Reviewer #3: Yes

Reviewer #4: Yes

4. Have the authors made all data underlying the findings in their manuscript fully available?

Reviewer #3: Yes

Reviewer #4: Yes

5. Is the manuscript presented in an intelligible fashion and written in standard English?

Reviewer #3: Yes

Reviewer #4: Yes

6. Review Comments to the Author

Reviewer #3: To authors,

The authors faithfully reacted to my comments, of which incorporation into the version markedly improved the paper quality. At last, what we learned here were 1) cesarean with labor may demand more opioid than that without labor (the fact), and 2) thus, labor + vs. – may be taken into consideration when deciding opioid-prescription-dose. Actually speaking, this does not much influence the daily obstetric practice; however, authors should not be blamed for this. I believe that postoperative pain is an important medical problem, and step-by-step clarification of each phenomenon may be of some use.

Reviewer #4: As I indicated previously, the methodology of the manuscript seems to be sound, and the authors responded appropriately to my questions.

7. PLOS authors have the option to publish the peer review history of their article (what does this mean?). If published, this will include your full peer review and any attached files.

Reviewer #3: No

Reviewer #4: No

---

## [Editor Report · Acceptance letter]

30 Jun 2021

PONE-D-20-40071R2 

Labor prior to cesarean delivery associated with higher post-discharge opioid consumption 

Dear Dr. Ende:

I'm pleased to inform you that your manuscript has been deemed suitable for publication in PLOS ONE. Congratulations! Your manuscript is now with our production department. 

Kind regards, 

on behalf of

Dr. Robert Jeenchen Chen 

Academic Editor

PLOS ONE